# Identifying key areas of post-secondary student stress: Principal component analysis of the Post-Secondary Student Stressors Index (PSSI)

**Danielle Schwartz**[1], **Essence Perera**[1,2], **Ian Clara**[2], **Brooke Linden**[3], **Shay-Lee Bolton**[1,2,4] *

1 Department of Psychiatry, University of Manitoba, Winnipeg, Manitoba, Canada, 2 Department of Community Health Sciences, University of Manitoba, Winnipeg, Manitoba, Canada, 3 Health Services and Policy Research Institute, Queen's University, Kingston, Ontario, Canada, 4 Department of Psychology, University of Manitoba, Winnipeg, Manitoba, Canada

* shay-lee.bolton@umanitoba.ca

**Data Availability Statement:** There are ethical restrictions to sharing this data, put in place by the 15 ethics review board(s) from which we received

## Abstract

Stress and mental health problems are prominent in Canadian postsecondary populations. Experiences of stress vary widely across the country due to student differences. The Post-Secondary Student Stressors Index (PSSI) is a 46-item inventory that assesses student-specific stressors. The PSSI has previously been validated in Ontario and Canada more broadly. However, as a new scale, the PSSI requires validation across different contexts. The purpose of this study was to use the PSSI to determine the areas of stress specific to the Manitoba population, utilizing more detailed information on severity and frequency of stress. Data were drawn from a Manitoba subset of the PSSI respondents. This resulted in a sample size of 2856 students from the University of Manitoba. Each item on the PSSI-46 was transformed into a new variable representing the inclusion of the frequency and severity variables for each stressor. Principal component analysis (PCA) was used to explore relationships between the 46 items of the PSSI. Direct oblimin rotation method was used to examine fit indicators. Spearman's rho was used to examine correlations between the revised PSSI score and those on other instruments. Cronbach's alpha was used to determine the internal reliability. The PCA produced 10 stress components consisting of 40 items. Six items did not load onto any components and were therefore excluded from component formation. The components demonstrated strong psychometric properties and internal validity. This study utilized the PSSI measure in a novel way, both through context and statistical assessment. These components of stress may be employed in future research that assesses post-secondary student stress.

## Introduction

Recent studies at Canadian universities indicate that student stress is increasing in prevalence [1]. Post-secondary students experience a wide range of chronic stressors in areas including

clearance. A stipulation of our ethics approvals across all sites was that the students' data would not be shared without an additional layer of ethics approval. All secondary analyses require a subsequent ethics request to the Queen's University's Health Sciences and Affiliated Teaching Hospitals Research Ethics Board (HSREB) as the primary ethics body that approved this study. Requests regarding access to the data used in this study can be directed to the HSREB (hsreb@queensu.ca, 613-533-6000 ext. 78223), who will forward these requests to the data custodian (Dr. Brooke Linden, brooke.linden@queensu.ca). Data are available upon request.

**Funding:** This study was financially supported by the Department of Psychiatry, University of Manitoba [https://umanitoba.ca] in the form of a grant (#326741) received by SLB. No additional external funding was received for this study.

**Competing interests:** I have read the journal's policy and the authors of this manuscript have the following competing interests: Dr. Brooke Linden is one of the original creators of the Post-Secondary Student Stressors Index. However she does not gain financially from any research done on the scale. This research project was designed independently by the senior author and any potential bias that may have been introduced by Dr. Linden's involvement was taken into consideration by the entire authorship group.

academics, campus environment, personal, and interpersonal [2]. Stress and ability to cope have implications for wellbeing. The stress sensitization model posits that chronic stress is associated with increases in mental health problems including mood disorders and substance use [3]. Evaluation of the different areas of stress in the post-secondary setting–to determine which areas are most impactful–will aid in the development of more effective resources to support student wellbeing.

Many studies indicate stress as a one-dimensional, broad phenomenon, with failure to distinguish between components of stress [4]. This approach limits the identification of areas of stress that are most relevant to post-secondary students. For example, it is important to make the distinction between the stress categories that are most relevant for general population samples and the stress categories that are most relevant for post-secondary students. Common areas of stress for general population adults include work, major life events, physical and mental health, interpersonal relationships, and finance [5]. Post-secondary students' experience of academic-related stressors, on top of these generalized stressors, can exacerbate stress-related symptoms [6]. It is important to understand the areas of stress that are most impactful for post-secondary students, and to recognize that these stressors are distinct from general population stressors. A failure to make this distinction will result in the identification and allocation of resources and supports that are not tailored to the needs of students.

Existing stress evaluation instruments include the Perceived Stress Scale (PSS-10) [7] and the Kessler Psychological Distress Scale (K10) [8], which are brief measures of non-specific, overall aspects of stress and psychological distress, respectively. These instruments have been largely designed for general adult populations. They offer no clarification around areas of stressors that are most relevant to specific subgroups of the population. Given the unique stressors experienced by post-secondary students, such stress measures may not be suitable. Other existing instruments that evaluate post-secondary student stress, including the National College Health Assessment [9], the Mental Health Continuum–Short Form [10], and the Canadian Campus Wellbeing Survey [11], have been suggested to be lacking in their validity for a number of reasons. Some of these include evaluation of only one dimension of stress, failure to involve a geographically or socioculturally diverse student sample, or being too broad or inclusive of stress-related items that are not applicable to the post-secondary student experience [4]. Gaining an understanding of different types of stressors is crucial in the management of stress [12]. In the context of post-secondary students, a component-specific approach to stress would offer more nuanced information for policymakers and mental health measures [12]. In addition, a clearer understanding of type, duration, and intensity of stressors could further elucidate targets for stress reduction in the post-secondary context.

The Post-Secondary Student Stressors Index (PSSI) [4] is a novel measure of student-specific stressors that captures stressors across five key domains (academic, learning environment, campus culture, interpersonal, and personal). During the 2020–2021 academic year, the PSSI was pilot tested amongst post-secondary students attending one of 15 Canadian post-secondary institutions. A recent paper [4] examined the PSSI's psychometric properties using an exploratory factor analysis (EFA) demonstrating moderate psychometric properties in a sample of post-secondary students (n = 535) at an Ontarian university. A large cross-national sample (n = 12,577) was used to extend these findings [6,13]. The cross-national study utilized a confirmatory factor analysis (CFA) to evaluate the internal structure of the PSSI.

Prior work on the PSSI has been limited in a number of ways. The original psychometric evaluations of the PSSI [4,6,13] did not make use of the Likert scale to include features of severity and frequency of stressors in the creation of the five domains. A dual-method of depicting both severity and frequency of stress minimizes data loss and produces a more detailed and accurate depiction of stressors–enabling the observation of priority areas for improvement.

Further, the component structure for the establishment of the five domains was not founded on psychometric evaluation of the data, but rather face validity–which introduces potential bias surrounding domain construction. Although the PSSI has been psychometrically evaluated previously, some aspects have not been rigorously examined (e.g., internal consistency, reliability analyses, dimensionality tests), while use of a relatively small sample from a single university indicates that results may not be generalizable to other settings [14,15]. In addition, differences exist across postsecondary student bodies across the country, leading to differing experiences of student stress in different regions [16]. The Manitoba sample utilized in this study is comprised of a larger, unique population of demographically varied postsecondary students, which will aid in testing and improving upon the generalizability of the PSSI.

It is important to utilize statistical methods to validate the PSSI in the Manitoba population–which has not been done previously. Prior work was based on an interpretation of an EFA structure, which provided differentiation between the factor loadings and the final domains presented [4]. However, principal component analysis (PCA) may be seen as a more accurate tool in the context of indices when performing exploratory work with the dimensions of frequency and severity of stress [17]. PCA, a data-driven approach to index analysis, is a powerful statistical tool for interpreting large datasets and improving their dimensionality while retaining as much information as possible [17]. This rigorous statistical assessment and the refinement of components of stress could improve upon the psychometrics, internal validity, and interpretability of the PSSI scale.

The present study aimed to create components of post-secondary student stress using PCA among a subsample of Manitoba post-secondary students (n = 2856) derived from the nationwide PSSI survey to extend findings from prior work. Clear subcategories of stressors will help to contextualize items in a coherent manner for future applicability, and improve the scale's interpretability and practicality. A secondary aim was to assess the construct validity of the resulting components from the PCA analysis.

## Methods

### Ethics statement

This research received ethics clearance from Queen's University's Health Sciences and Affiliated Teaching Hospitals Research Ethics Board, as well as the University of Manitoba's Health Research Ethics Board (HS24034 [H2020:295]). All participants were provided with a letter of information and indicated their consent digitally before being granted access to the survey questions.

### Sample

The original pan-Canadian data was collected across three timepoints via online surveys over the course of the 2020–2021 academic year [6,13]. The purpose of the survey was to collect data on university student stress and mental health outcomes. T1 occurred in October 2020 (n = 4954), T2 in January 2021 (n = 4576), and T3 in late March/early April 2021 (n = 3083). Demographic characteristics were similar across all time points. Respondents were predominantly female with a mean age ranging from 23.4–24.9. Further detail related to study design can be found elsewhere [2].

This paper used a subset of the pan-Canadian survey, including only respondents who attended a Manitoba post-secondary institution (T1 included 972 students, T2 included 867 students, and T3 included 606 students). The design of the study was a secondary data analysis of a longitudinal cohort. Separate datasets across all three timepoints were merged into one database. The pan-Canadian survey created a respondent identifier (ID) based on the

participant initials and their year of birth. Due to the format of this respondent ID, once the data were merged, there were multiple duplicate IDs at the same time point (i.e., the unique ID was not completely unique across individuals). As such, a list of duplicates was created, and five 'stable' demographic characteristics were examined (ID, sex, birth year, international student, first generation student). For cases in which there were multiple individuals with the same unique ID across multiple timepoints, if they did not match on all five stable characteristics in the same timepoint, we assumed that these were different individuals and assigned a unique ID. If individuals matched on all characteristics, then these cases were assumed to be the same individual. This resulted in a total sample size of n = 2856 post-secondary individuals from Manitoba. All analyses were conducted using the first timepoint that an individual completed the survey, which may have been at T1, T2 or T3.

## Measures

**Post-Secondary Student Stressors Index (PSSI).**   The PSSI assessed 46 stressors, which included items focused on academics (e.g., "Heavily weighted assignments"), learning environment (e.g., "Poor communication from professor"), campus culture (e.g., "Adjusting to my program"), interpersonal stressors (e.g., "Balancing a social life with academics"), and personal stressors (e.g., "Balancing working at my job with my academics"). Respondents were asked to rank the severity of stress on a Likert scale that was associated with each item ("How stressful is this?"), as well as the frequency with which the stress occurred ("How often are you stressed about this?"). Response options for severity were represented on a scale from 0 ('N/A' or 'didn't happen') to 4 ('extremely stressful'). Response options for frequency were represented on a scale from 0 ('N/A' or 'never') to 4 ('almost always'). Further details on the psychometric properties of the PSSI have been published elsewhere [2]. Each item on the PSSI was transformed into a new variable that represented the multiplication of the frequency and severity variables for that item. This approach allowed for the capture of both whether a stressful situation occurred, how many times, and its level of severity of stress for the respondent. Higher scores indicated higher stress levels and impact.

## Analysis

All analyses were performed in SPSS Statistics, version 28.0. First, descriptive statistics were calculated to describe the local sample. Next, principal component analysis (PCA) was used to explore the relationship between the 46 items of the PSSI. The direct oblimin rotation method was used to examine fit indicators, including KMO and Bartlett's test of sphericity, scree plot, and the suppression of coefficients below 0.4 for inclusion of an item on a given dimension. Rotated factor loadings and item content for each extracted component were subsequently inspected for interpretability. Cronbach's alpha was used to determine the internal reliability of the revised structure.

# Results

Demographics of the sample population are included in Table 1. The sample was primarily female (71.0%), with a mean age of 23.0 years (SD = 0.1). The majority of the sample was studying at the undergraduate level (81.3%), with first year undergraduates representing the largest subgroup (28.5%). Roughly 16% (n = 477) of the sample was international students, whilst 25.4% (n = 721) of the sample were first-generation students.

Results of the PCA produced 10 stress components consisting of 40 individual stressor items, shown in Table 2. Component 1, labeled *Academic Comparison*, consisted of nine items describing social comparison with peers and personal achievement in an academic

**Table 1. Demographic characteristics of the Manitoba PSSI sample.**

| | Mean (SE) | |
|---|---|---|
| Age | 23.0 (0.12) | |
| | **N** | **%** |
| Sex | | |
| Male | 779 | 27.5 |
| Female | 2010 | 71.0 |
| Non-binary | 40 | 1.4 |
| Relationship status | | |
| Single | 1540 | 54.6 |
| Partnered | 1281 | 45.4 |
| Children | | |
| Yes | 229 | 8.1 |
| No | 2615 | 91.9 |
| Residence | | |
| On campus | 80 | 2.9 |
| Off campus | 2727 | 97.1 |
| Level of study | | |
| 1st year undergraduate | 810 | 28.5 |
| 2nd year undergraduate | 547 | 19.3 |
| 3rd year undergraduate | 447 | 15.7 |
| 4th year undergraduate or higher | 505 | 17.8 |
| Graduate/ professional program | 443 | 15.6 |
| Other (Please specify) | 89 | 3.1 |
| | **N** | **%** |
| Student status | | |
| Full-time student | 2542 | 89.9 |
| Part-time student | 286 | 10.1 |
| International student | | |
| No | 2375 | 83.3 |
| Yes | 477 | 16.7 |
| First generation student | | |
| No | 2118 | 74.6 |
| Yes | 721 | 25.4 |
| Student athlete | | |
| No | 2271 | 79.5 |
| Yes | 476 | 16.7 |
| Prefer not to answer | 108 | 3.8 |
| Area of study | | |
| General arts and education | 855 | 30.0 |
| Sciences | 1200 | 42.2 |
| Engineering and applied sciences | 245 | 8.6 |
| Law, business and policy studies | 306 | 10.8 |
| Other | 240 | 8.4 |
| GPA | | |
| 80–100% / A | 1571 | 55.1 |
| 70–79% / B | 865 | 30.3 |
| 60–69% / C or lower | 224 | 7.9 |
| Prefer not to answer | 193 | 6.8 |

**Table 2. Principal component analysis demonstrating PSSI items, by component.**

| Component | Items | Component Loadings |
|---|---|---|
| *Academic Comparison* | a) Academic competition among my peers | .427 |
| | b) Feeling like I'm not working hard enough | .613 |
| | c) Feeling like my peers are smarter than I am | .641 |
| | d) Pressure to succeed | .662 |
| | e) Comparing myself to others | .668 |
| | f) Comparing my life to others' on social media | .430 |
| | g) Meeting people's expectations of me | .592 |
| | h) Meeting my own expectations | .595 |
| | i) Worrying about getting into a new program after graduating | .434 |
| *Thesis Achievement with Supervisor* | a) Working on thesis | .861 |
| | b) Performing well at my professional placement (i.e., practicum, clerkship, etc.) | .563 |
| | c) Meeting my thesis or placement supervisor's expectations | .914 |
| | d) Lack of mentoring from my thesis or placement supervisor | .841 |
| *Exam-related Stress* | a) Preparing for exams | -.712 |
| | b) Writing exams | -.805 |
| | c) Writing multiple exams on the same day | -.841 |
| | d) Exams worth 50% or more of the final grade | -.836 |
| | e) Receiving a bad grade | -.526 |
| *Healthy Lifestyle* | a) Making sure that I get enough sleep | .675 |
| | b) Making sure that I get enough exercise | .820 |
| | c) Making sure that I eat healthy | .862 |
| | d) Having to prepare meals for myself | .724 |
| | e) Balancing working at my job with my academics | .438 |
| | f) Balancing my extracurriculars with academics | .462 |
| *Finance* | a) Having to take student loans | .946 |
| | b) Working on paying off debt | .952 |
| *Professor/Advisor Interaction* | a) Poor communication from professor | -.903 |
| | b) Unclear expectations from professor | -.886 |
| | c) Lack of guidance from professor | -.833 |
| *Interpersonal Relationships* | a) Making new friends | .883 |
| | b) Maintaining friendships | .810 |
| | c) Networking with the "right people" | .725 |
| | d) Feeling pressure to socialize | .761 |
| *Harassment/Discrimination* | a) Discrimination on campus | .853 |
| | b) Sexual harassment on campus | .909 |
| *Academic Adjustment* | a) Adjusting to the post-secondary lifestyle | -.544 |
| | b) Adjusting to my program | -.637 |
| *Academic Workload* | a) Heavily weighted assignments | .575 |
| | b) Having multiple assignments due around the same time | .667 |
| | c) Managing my academic workload | .523 |

*The following items didn't load above the 0.4 threshold*: maintaining a high GPA, meeting with my professor, balancing a social life with academics, feeling guilty about taking time for my hobbies/interests, worrying about getting a job after graduating, worrying about reaching major "life events" (i.e., buying a house, marriage, children)

environment. Component 2, *Thesis Achievement with Supervisor*, was comprised of four items that highlighted career-oriented actions with the help of a supervisor, specifically in the furthering of a thesis assignment. Component 3, labeled *Exam-related Stress*, consisted of five items describing academic achievement, with a particular focus on the stress associated with

**Table 3. Reliability coefficients for the 10 components.**

| Component | Cronbach's Alpha |
|---|---|
| *Academic Comparison* | 0.89 |
| *Thesis Achievement with Supervisor* | 0.82 |
| *Exam-related Stress* | 0.87 |
| *Healthy Lifestyle* | 0.82 |
| *Finance* | 0.90 |
| *Professor/Advisor Interaction* | 0.89 |
| *Interpersonal Relationships* | 0.86 |
| *Harassment/Discrimination* | 0.72 |
| *Academic Adjustment* | 0.77 |
| *Academic Workload* | **0.75** |

exams. Component 4, *Healthy Lifestyle*, consisted of six items that described components of a healthy lifestyle and self-care, including adequate sleep and regular exercise. Component 5, *Finance*, described stress associated with student loans and debt. Component 6, *Professor/Advisor Interaction*, consisted of three items that described negative academic interactions with a professor. Component 7, *Interpersonal Relationships*, was comprised of four items that related to interpersonal relationships, including friendships and networking in a social environment. Component 8 assessed sexual discrimination and harassment and was labeled *Harassment/Discrimination*. Component 9, *Academic Adjustment*, described adjusting to one's academic program and corresponding lifestyle. Component 10 described stress related to assignments and assignment workload and was labeled *Academic Workload*.

Six items of the PSSI did not load onto any components (i.e., factor loading <0.4) and were therefore excluded from further analysis. These included: maintaining a high GPA, meeting with my professor, balancing a social life with academics, feeling guilty about taking time for my hobbies/interests, worrying about getting a job after graduating, and worrying about reaching major "life events" (i.e., buying a house, marriage, children).

The reliability of the total score of the PSSI was evaluated in the Manitoba sample ($\alpha$ = 0.95). Table 3 displays reliability coefficients (Cronbach's alpha) for each component of the PSSI. Cronbach's alpha values for each component fell in a relatively narrow range, from 0.73–0.90. Overall, each of the components exhibited moderate to high internal reliability, indicating that the items that make up each component are correlated with each other.

## Discussion

The present study set forth to create statistically-derived subcomponents of the PSSI–an instrument composed of 46 stressors within the post-secondary setting. The original pan-Canadian survey incorporated 15 institutions across Canada, while our study specifically focused on a subset of post-secondary students (n = 2856). The purpose of this research was to create ways to detect differences in stressors of post-secondary student populations through identification of specific aspects or "components" of stress that allow us to contextualize stress in a novel and meaningful manner.

The present research used PCA–a statistically-founded analysis framework–to establish components of stressors and distinct item loadings. The PCA determined which items created the foundation for each component of stress (and hence were a good fit), and which items were not relevant (and were thus dropped from further analysis). Results of the PCA established 40 items contextualized across 10 components. Psychometric properties of the

components were also evaluated, indicating moderate to high reliability of the items within their component framework.

The primary purpose was to utilize PCA to reorganize the PSSI items in a way that increased its interpretability whilst simultaneously minimizing the loss of information. This was done through the inclusion of both frequency and severity of each stressor item, which had not been done previously. This rigorous statistical assessment supported our creation of components of stress. Items that grouped together based on the PCA displayed face validity as well as good statistical fit. This statistical validation improves upon the psychometrics and interpretability of the PSSI scale and improves applicability in future research and policy decision.

Another means by which the components build upon the PSSI is in their utilization of the full Likert scale of measurement. The Likert scale is an important piece of the PSSI that was not utilized to its full potential in the original assessment. Likert scales are intended to represent items on a continuous, linear scale (i.e., a range from 0 ('N/A' or 'didn't happen') to 4 ('extremely stressful')). While the PSSI did collect information on the severity and frequency of a stressor, prior analysis collapsed severity and frequency of each item into a single binary variable (*"did you experience the stressor or not?"*). The present work made use of the Likert scale in the creation of the 10 components by performing a multiplication of the frequency and severity variables for each stressor on the PSSI. This approach allowed us to utilize the PSSI as it was originally intended–to depict stressors on a range of both severity (how impactful the stressor is) and frequency (how often the stressor is experienced). In doing so, we minimized the loss of data and variance and produced a more detailed depiction of priority areas of student-specific stressors.

While the current study offers many strengths in its application for future research–including a more rigorous statistical foundation, clearer conceptualization of priority areas of stress for post-secondary students, and a novel contextual population–some limitations are present. First, the current study was analyzed using one subset of the national data collection, specifically from Manitoba (n = 2856). As such, generalizability of our study results across Canada is uncertain, as the stressors experienced by Manitoba students may be distinct and unique from the stressors experienced by students living in other regions of Canada. Future studies should assess the validity of the identified components in varied student samples across other Canadian regions. Secondly, the relationship of stress with other mental health issues, such as anxiety and depression, was not explored by this study. Exploration of the association between mental health symptoms and stress will allow researchers to paint a more holistic image of the experience of stress in post-secondary populations. Finally, data used in these analyses could have been collected at any of the three timepoints (T1, T2, T3) included in the study. There are several reasons why students might have missed responding to a particular timepoint; uncertainty surrounding these reasons is an inherent limitation of the data and may have led to systematic bias in the data. However, capturing data for as many students as possible across timepoints of measurement enabled increased sample size, variability in stressor experiences throughout the academic year, and potential to capture diverse stressors.

Our results highlight specific components of stress that are common among post-secondary students that could be utilized in future research. The practicality of the components lie in their identity as a statistically sound, clearly-defined variation of the PSSI that can be used to identify priority areas of stress for post-secondary students. Future studies can utilize the components to identify the distribution of stressors in unique subpopulations (e.g., international students). Comprehensive downstream approaches to post-secondary students' mental health begin with a fundamental understanding of key areas of stress that are affecting students [18].

Thus, the organization of the PSSI into a statistically-grounded framework for analysis will allow for its use in future research and policy change.

## Acknowledgments

The authors would like to thank Dr. Alyson Mahar for her contribution to this work.

## Author Contributions

**Conceptualization:** Danielle Schwartz, Essence Perera, Ian Clara, Brooke Linden, Shay-Lee Bolton.

**Data curation:** Shay-Lee Bolton.

**Formal analysis:** Danielle Schwartz, Essence Perera, Shay-Lee Bolton.

**Funding acquisition:** Shay-Lee Bolton.

**Methodology:** Essence Perera, Ian Clara, Brooke Linden, Shay-Lee Bolton.

**Project administration:** Shay-Lee Bolton.

**Software:** Shay-Lee Bolton.

**Supervision:** Essence Perera, Ian Clara, Shay-Lee Bolton.

**Writing – original draft:** Danielle Schwartz, Essence Perera, Shay-Lee Bolton.

**Writing – review & editing:** Danielle Schwartz, Essence Perera, Ian Clara, Brooke Linden, Shay-Lee Bolton.

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
