## [Decision Letter · Decision Letter 0]

9 Sep 2024

PMEN-D-24-00154

Identifying key areas of post-secondary student stress: Model reduction using principal component analysis in the Post-Secondary Student Stressors Index (PSSI)

PLOS Mental Health

Dear Dr. Bolton,

Thank you for submitting your manuscript to PLOS Mental Health. We apologise for the severe delay in reaching a decision on your manuscript and thank you for your understanding and patience. After careful consideration and taking the reviewer comments into consideration, we feel that it has merit but does not fully meet PLOS Mental Health’s publication criteria as it currently stands. Therefore, we invite you to submit a revised version of the manuscript that addresses the points raised during the review process.

Please ensure you address all comments raised by reviewers (which you can see in full below). Please pay particular attention to the points raised by reviewer 3.

We look forward to receiving your revised manuscript.

Kind regards,

Karli Montague-Cardoso

Executive Editor

PLOS Mental Health

Journal Requirements:

1. Thank you for your response to our query. You may please add Dr Maher in the acknowledgements. Thank you for your attention.

2. Please send a completed 'Competing Interests' statement, including any COIs declared by your co-authors. If you have no competing interests to declare, please state "The authors have declared that no competing interests exist". Otherwise please declare all competing interests beginning with the statement "I have read the journal's policy and the authors of this manuscript have the following competing interests:"

3. Please provide a/amend your detailed Financial Disclosure statement. This is published with the article. It must therefore be completed in full sentences and contain the exact wording you wish to be published.

**Please only choose the relevant sentences from below**

1. Please clarify all sources of funding (financial or material support) for your study. List the grants (with grant number) or organizations (with url) that supported your study, including funding received from your institution. 

2. State the initials, alongside each funding source, of each author to receive each grant.

3. State what role the funders took in the study. If the funders had no role in your study, please state: “The funders had no role in study design, data collection and analysis, decision to publish, or preparation of the manuscript.”

4. If any authors received a salary from any of your funders, please state which authors and which funders.

4. "H2020 295 Approval Cert.pdf" is currently uploaded as an 'Supporting Information' file type. Please ensure that it is uploaded as 'Other'.

5. We note that you have indicated that there are restrictions to data sharing for this study. For studies involving human research participant data or other sensitive data, we encourage authors to share de-identified or anonymized data. However, when data cannot be publicly shared for ethical reasons, we allow authors to make their data sets available upon request. For information on unacceptable data access restrictions, please see http://journals.plos.org/plosone/s/data-availability#loc-unacceptable-data-access-restrictions. 

Additional Editor Comments (if provided):

Reviewers' comments:

Reviewer's Responses to Questions

**Comments to the Author**

1. Does this manuscript meet PLOS Mental Health’s publication criteria? Is the manuscript technically sound, and do the data support the conclusions? The manuscript must describe methodologically and ethically rigorous research with conclusions that are appropriately drawn based on the data presented.

Reviewer #1: Yes

Reviewer #2: Yes

Reviewer #3: No

2. Has the statistical analysis been performed appropriately and rigorously?

Reviewer #1: Yes

Reviewer #2: Yes

Reviewer #3: No

3. Have the authors made all data underlying the findings in their manuscript fully available (please refer to the Data Availability Statement at the start of the manuscript PDF file)?

Reviewer #1: Yes

Reviewer #2: No

Reviewer #3: Yes

4. Is the manuscript presented in an intelligible fashion and written in standard English?

Reviewer #1: Yes

Reviewer #2: Yes

Reviewer #3: Yes

5. Review Comments to the Author

Reviewer #1: • Lack of theoretical aspects

• Lack of recent scientific study.

• Rationale of current study is need.

• Scope of current study should be included.

• Significance of current study is lacking.

• Need to mention the design of the current study.

• Lack of information regarding study area.

• Need to add the Cronbach's alpha of each scale.

• Poor future directions.

• Limitations should be clear.

• Lack of conclusion.

• The relevance of Model reduction using principal component analysis is lacking.

Reviewer #2: Your manuscript is methodologically sound and makes a significant contribution to the field. Conduct a thorough proofreading to catch any minor typographical or grammatical errors that may have been overlooked. This will help ensure the manuscript meets the high standards expected by PLOS Mental Health.

Reviewer #3: This work left me with far more questions than answers. I’m listing my questions here roughly in order from those I feel are most significant to those that are less important (and might even become non-issues, depending upon how the earlier questions are resolved).

1) What is the intent of this work? The title and some content in the introduction suggest the primary aim is to reduce the number of items on the PSSI, but the methodology that is implemented is poorly suited to this goal. As a result, the work ends up suggesting that the domains underlying the PSSI should be revised.

2) Is there confidence in the measurement model underlying the PPSI-46? If so, a confirmatory methodology (such as formative structural equation modeling) could be used to identify which items to drop. This approach would enable direct evaluation of the impact of these changes on the overall goodness of fit, and would avoid the problem of the authors’ measurement model yielding different domains than have been previously reported in the literature.

3) What should readers conclude from the different domains that were found in previous analyses of the PSSI-46 (and the Brief-PSSI) and the findings reported in the present work? Do the discrepant findings result from differences in the sample, or from scoring differences (using Likert scales instead of binary variables to score the PSSI)? Which domains provide a better description of the PSSI measurement model, and what evidence supports this conclusion?

4) Which methods were used to identify the five domains in the PSSI-46, and how do these differ from the methods in the present work? What methods were used for factor extraction and rotation during development of the PSSI-46? What factor loading cutoff was used to retain items? What is the available psychometric evidence for the reliability and validity of the PSSI-46? Most of these details seem to be absent from the references cited by the authors but are important for contextualizing the present work. For example, researchers often use principal component analysis (PCA) to extract factors but describe this as exploratory factory analysis. Was this the approach used for the PSSI-46? If so, the lengthy justification for using PCA in the present work seems inappropriate.

5) The authors argue for the need to assess “different types of stress” (line 49) and criticize other works because they evaluate “only one dimension of stress” (line 54). What is the justification for assessing multiple types of stress? I can think of several reasons but am interested in the authors’ motivations here. If it is indeed important to measure several dimensions of stress, why do the authors derive a 10-domain measure of stress but only report statistics relating to a single, overall stress measure?

6) What evidence is there to support the reliability of the PSSI-40? Internal validity isn’t reported (although lines 292-295 implies that it is). Similarly, lines 295-296 suggest test-retest reliability was calculated, but no details about this are provided in the paper.

7) Should readers be surprised that the PPSI-40 “achieved similar psychometric properties in comparison with the PSSI-46” (lines 275-276), given that only total scores were used and the two instruments share 40 items?

8) How do the authors conceptualize “diversity”? Does this term – which is mentioned several times – refer to diversity in geographic location, socioeconomic status, first generation status, gender, race, ethnicity, native language, or … ? Lines 80-81 seem to imply that the actual concern is not really diversity, but is instead that findings derived from administering the PSSI in other geographic locations may not be generalizable to students in Manitoba. If so, this should be stated explicitly and comparisons should be made between Manitoba findings and findings from other locations. Otherwise, more information should be provided about what "diversity" means in the present context. This could require revising Table 1 to characterize the diversity represented in the sample (to the extent that this information is available). One could also argue that descriptive statistics should be reported for PSSI (and PSS-10, K10, and CD-RISC) scores disaggregated by the diversity characteristic(s) of interest.

9) How many duplicates were in the original data, requiring the application of the steps described in lines 124-129? Presumably there would be no concern regarding duplicates if only the T1 data were examined. What sample size would have resulted from including only T1 data, and why did the authors choose instead to include data from other waves? Might this practice have added undesired variance (and, perhaps, systematic biases of some sort) to the data? What might lead a student to miss T1 but provide responses to T2 or T3?

10) Is the reduction in items from 46 to 40 meaningful from the perspective of the respondent? What evidence or logic supports this position? Is either version superior to the Brief-PSSI (which contains only 10 items), and why?

11) What are the samples sizes for each wave in Manitoba? Lines 107-108 seem to report sample sizes for all of Canada.

In light of these unanswered questions, substantial revisions are required before this work can be considered for publication. These revisions could possibly include major changes to the methodology.

6. PLOS authors have the option to publish the peer review history of their article (what does this mean?). If published, this will include your full peer review and any attached files.

**Do you want your identity to be public for this peer review?** For information about this choice, including consent withdrawal, please see our Privacy Policy.

Reviewer #1: No

Reviewer #2: **Yes: **Charles Ganaprakasam

Reviewer #3: No

---

## [Decision Letter · Decision Letter 1]

17 Dec 2024

Identifying key areas of post-secondary student stress: Principal component analysis of the Post-Secondary Student Stressors Index (PSSI)

PMEN-D-24-00154R1

Dear Dr. Bolton,

We are pleased to inform you that your manuscript 'Identifying key areas of post-secondary student stress: Principal component analysis of the Post-Secondary Student Stressors Index (PSSI)' has been provisionally accepted for publication in PLOS Mental Health.

Best regards,

Karli Montague-Cardoso

Staff Editor

PLOS Mental Health

Reviewer Comments (if any, and for reference):

Reviewer's Responses to Questions

**Comments to the Author**

1. If the authors have adequately addressed your comments raised in a previous round of review and you feel that this manuscript is now acceptable for publication, you may indicate that here to bypass the “Comments to the Author” section, enter your conflict of interest statement in the “Confidential to Editor” section, and submit your "Accept" recommendation.

Reviewer #1: All comments have been addressed

Reviewer #3: All comments have been addressed

2. Does this manuscript meet PLOS Mental Health’s publication criteria? Is the manuscript technically sound, and do the data support the conclusions? The manuscript must describe methodologically and ethically rigorous research with conclusions that are appropriately drawn based on the data presented.

Reviewer #1: Yes

Reviewer #3: Yes

3. Has the statistical analysis been performed appropriately and rigorously?

Reviewer #1: Yes

Reviewer #3: Yes

4. Have the authors made all data underlying the findings in their manuscript fully available (please refer to the Data Availability Statement at the start of the manuscript PDF file)?

Reviewer #1: Yes

Reviewer #3: Yes

5. Is the manuscript presented in an intelligible fashion and written in standard English?

Reviewer #1: Yes

Reviewer #3: Yes

6. Review Comments to the Author

Reviewer #1: Good work

Reviewer #3: This revision is a very strong manuscript, with a clear focus and a concise presentation. It addresses all of my concerns with the prior submission, as well as those raised by the other reviewers. Very impressive work!

Just one extremely minor niggle: Line 234 is missing an open parenthesis.

7. PLOS authors have the option to publish the peer review history of their article (what does this mean?). If published, this will include your full peer review and any attached files.

**Do you want your identity to be public for this peer review?** For information about this choice, including consent withdrawal, please see our Privacy Policy.

Reviewer #1: No

Reviewer #3: No
